# An Exploration of the Psycho-Social Benefits of Providing Sponsorship and Supporting Others in Traditional 12 Step, Self-Help Groups

**DOI:** 10.3390/ijerph18052208

**Published:** 2021-02-24

**Authors:** William McGovern, Michelle Addison, Ruth McGovern

**Affiliations:** 1Department of Social Work Education and Community Wellbeing, Faculty of Health and the Life Sciences, Northumbria University, Newcastle upon Tyne NE7 7XA, UK; 2Criminology Department, Faculty of Arts Design and Social Sciences, Northumbria University, Newcastle upon Tyne NE7 7XA, UK; michelle.addison@northumbria.ac.uk; 3Population Health Sciences Institute, Newcastle University, Newcastle upon Tyne NE2 4AX, UK; r.mcgovern@ncl.ac.uk

**Keywords:** self-help, sponsorship, recovery, helping others, psycho-social benefits

## Abstract

Sponsorship is a key feature of traditional drug and alcohol self-help groups. It is a source of interpersonal support provided by an individual who is in a more advanced stage of recovery to an individual at an earlier stage of recovery. Whilst it is widely recognised that sponsorship is beneficial to the person receiving it, little is known about the psychological and social benefits that sponsors derive from providing sponsorship to others. We conducted in-depth qualitative interviews with 36 long-term self-help users (6 months−10 years) with experience of sponsoring the recovery of others, recruited from three traditional types of self-help groups in the North of England. Interviews examined sponsors’ experiences of providing sponsorship within their own recovery process. Sponsors reported that providing sponsorship to others increased their own self-awareness, social skills, and social competence when it came to engaging with others. In addition, sponsors derived an increased sense of psychological wellbeing and positive social approval from helping others. Over the longer term, sponsorship becomes a meaningful and purposeful activity as it allows those providing it to be productive, make meaning and maintain a non-addicted identity. Additionally, sponsorship is a process which is beneficial for those who have little access to wider social networks.

## 1. Introduction

Sponsoring is a specific form of helping relationship which is often associated with traditional types of 12 step groups such as Alcoholic Anonymous (AA) and Narcotics Anonymous (NA) [1]. It is best described as an open-ended period of one-to-one tutelage in which a more experienced member provides less experienced members with practical support and spiritual guidance away from the self-help group setting [2]. Because of the focus of the sponsoring relationship, sponsorship is seen as being key to the continuation of groups and their functioning over time [3]. Whilst the actual term “sponsorship” was not included in the drafting of the original 12 steps philosophy, the idea of helping others is a key part of its values and practices [4]. Step 12 of the programme encourages group members to provide active support to others, wherein those who “*[have] had a spiritual awakening as a result of these steps, [try] to carry this message to addicts, and to practice these principles in all our affairs*” [5]. Within this sense, supporting the recovery of others is a key stage of the sponsor’s own recovery programme.

There are a small number of studies that have explored the ways in which those receiving the intervention of sponsorship benefit from their involvement in the sponsoring process. Project MATCH which included 1726 self-help participants from inpatient and community-based settings identified that recipients of sponsoring were more likely to be progressing in their recovery and abstaining from alcohol than those who were not being supported by a sponsor [6]. In this study, it was identified that those receiving sponsorship benefitted from the guidance provided by sponsors, encouragement, support around personal challenges and from the ways their sponsors reinforced AA teachings and principles. In a similar study which involved (*n* = 256) previously alcohol-dependent participants, [7] were able to show that having a sponsor within the early stages of an AA programme was found to be predictive of abstinence from alcohol, and other drugs such as cannabis and cocaine. Self-help groups such as NA and AA do recognise that sponsorship can be problematic for the person receiving it. Concerns are associated with the vulnerability of being a newer member (emotional or physical dysphoria) and because there is no formal outline of the specific duties a sponsor is expected to provide [3]. Sponsoring can therefore encompass a wide range of different activities and groups such as AA and NA have begun to provide guidance to newcomers on the types of characteristics they should look out for when considering a sponsor [8].

In addition to the benefits gained by the person receiving sponsorship, providing sponsorship also is often seen as key to the sponsor maintaining their own recovery. Commonly used group slogans remind sponsors that to keep it, they must also “give it away” [9] (p. 142). Whilst the concept of recovery is contested and elusive [10], it is often used synonymously with abstinence and “sobriety” (Betty Ford Institute Consensus Panel, 2007, p. 222). However, a conceptual framework developed from a systematic review of recovery literature identified a number of essential elements beyond abstinence which underpin the recovery process [11]. Using the acronym CHIME, these are explained as: connectedness; hope and optimism about the future; identity; meaning in life and empowerment [11]. More recently, there has been a wider recognition of other social and psychological factors that are important to an individual’s recovery [10,11,12], wherein psychological wellbeing and social connectedness are acknowledged as central aspects of the recovery process [13]. As such, there is now general agreement among scholars and empirical theorists that the process of recovery is a socially mediated and multidimensional process which requires ongoing activity and action by the individual [11,13].

Research into self-help and mutual-aid programmes for a wide range of health and social issues has illustrated the reciprocal benefit brought about by group-based, peer-led programmes of support. These studies have reported that an increased sense of psychological wellbeing is experienced by the ‘helper’, when it is fed back to them that their support has been experienced as helpful by another person [14], for example, the “helper therapy principle” [15], “the helpers high” [16] and “the helper’s role” [17]. Within NA and AA programmes, providing sponsorship to others has been found to be strongly associated with improved and sustained abstinent rates in the person providing the sponsorship after one year [18]. Longitudinal studies’ have shown that 90% of those who go on to longer-term abstinence from substance use (5–7 years) will have been a sponsor at some point [19]. However, as these studies do not examine the causation relationship between sponsorship and abstinence, it is unclear whether sponsorship increases abstinence rates, or conversely whether it is sustained abstinence which increases the likelihood of progressing to sponsoring others. Despite this limitation, providing sponsorship to others is often perceived in empirical research as being important in the individual’s own recovery [18,19,20].

A small number of theorists have explored the concept of sponsorship in self-help groups and identified that sponsors have elevated status as “senior members” [3]. Sponsors are known for having “expertise” [21] in self-help processes, helping others and the resolution of their own substance-related concerns. However, to our knowledge, no research has examined these concepts in detail, or the wider social and psychological benefits derived from providing sponsorship or considered the potential impacts of providing it for the sponsor. Providing sponsorship can be a personally challenging process for the sponsor as it requires the individual to share their own personal experiences and navigate social interactions with others in early stages of recovery. In this context, it has been identified that sponsors can risk their own recovery (relapse) by taking on too much responsibility for others and their recovery [8]. The following research was developed to explore how sponsors experienced the process of providing sponsorship and to identify the psychological and social impact of providing sponsorship to others. It is concerned with identifying and discussing the ways in which these social and psychological factors mediated the sponsors own recovery. Sponsorships draw a range of benefits from supporting others, this includes, increased self-awareness and self-understanding, increased social competence, positive social approval and psychological wellbeing. Over the longer term, sponsorship provides opportunities for those providing it to be productive, make meaning and maintain a non-addicted identity.

### 1.1. Study Design

This study utilised a qualitative method, collecting in-depth data from face-to-face interviews. This approach was used to facilitate the disclosure of sensitive information [22] and give voice to the lived experience of the participants [23,24,25]. A thematic topic guide which contained a set of standard questions and prompts to facilitate discussion was developed and informed the basis for semi structured interviews. The topic guide contained questions which related to the following areas: group history and involvement, sponsorship provision in group, motivations to sponsor, experiences of sponsorship, perceived benefits and actual impacts, level of planned involvement in sponsorship over time. This flexible format provided participants with an opportunity to discuss matters of importance to them whilst also allowing for comparison across the sample. Previous fieldwork has shown that self-help groups are often experienced as hard to access by parties outside of the group. As such, the lead researcher (WM) was required to first build trust and relationships with established, long-standing members. This approach resulted in access being “gifted” to networks and connections wherein these longstanding members “vouchsafed” for the researcher, prior to conducting interviews [26].

On introduction to the groups, presentations were given at a regional service user group and at a number of open NA and AA meetings. Purposive and snowball sampling strategies were utilised [27]. Snowballing was a particularly useful strategy to access this hard-to-reach group as users were recruited from within their own networks [28,29]. Interested participants were asked to contact a dedicated telephone number, at which point we confirmed their eligibility and arranged a face-to-face interview. To be eligible to take part participants had to be drug free for a minimum of six months prior to this study and be a current or ex member of a self-help group. They were also required to have been a sponsor at some point during their involvement with self-help.

### 1.2. Participants

In total, 36 participants were recruited into this study from three traditional (NA and AA) 12 step self-help groups. Participants described themselves as having previously been primary alcohol users (*n* = 9) and primary heroin/crack users (*n* = 27). All of the participants had been involved in poly drug/alcohol use and accessed some sort of structured treatment prior to their involvement in self-help groups. However, many were not in contact with structured treatment services at the point of entering self-help. Recruitment continued until it was agreed by the research group that data saturation had been reached; defined as no new themes emerged in three consecutive interviews. The resulting study sample consisted of *n* = 24 male and *n* = 12 female participants. Of these *n* = 34 assigned themselves to White/British and *n* = 2 to White/Irish. The ages of participants ranged from 24–52 years. At the point of interview, most participants (*n* = 33) were actively sponsoring another user; most (*n* = 28) had sponsored more than one other individual in the past. The minimum period any one participant had been sponsoring was three months and the maximum was 10 years: the average time spent sponsoring others was approximately 3–4 years. Whilst most of these sponsoring relationships were within the community, two participants were also sponsoring individuals by exchanging letters in a prison setting.

### 1.3. Analysis

All interviews were audio recorded and transcribed verbatim. The lead researcher (W.M.) checked the transcripts for accuracy and read in detail to become familiarised with the data. Descriptive and analytical codes were extracted from the data (W.M.) which related to understanding the activities sponsors undertook in providing their duties and then in relation to the social and psychological benefits that users claimed or inferred they derived from the process of sponsorship. Examples of initial codes included the importance of belonging, social acceptance, developing competence in self-help, hope for the future, knowledge and skill development, sponsors as role models, and the importance of participation. As this study progressed, the codes were refined and subsequently explored during interviews with respondents and applied to all transcripts. These refined codes included the importance of self-understanding, social status, social processes, psychological wellbeing, increased meaning and purpose, the importance of being productive and making an active contribution. The lead researcher (W.M.) coded all transcripts. Additionally, a subsample of eight interviews were independently second coded (R.M., M.A.), then compared and discussed in analysis meetings. Those members of the research team facilitating the secondary coding were given an audio recording of interviews, a word version of each of the transcript and blank excel sheet which contained all of the existing codes. No new codes were added or deemed necessary at the conclusion of the secondary coding phase. In addition to this and after in-depth discussion, there was universal agreement among all three researchers in relation to the primary themes and sub codes.

## 2. Results

In this section, we present the main primary themes and sub themes that were extracted from the data in relation to the study aims and objectives: increased self-awareness and self-understanding; increased social competence and wellbeing; meaning making and psychological wellbeing.

Increased self-awareness and self-understanding were mentioned and discussed in almost every interview—this theme was derived from and included a number of sub themes which were labelled: the importance of engaging with others; learning from others; exchanging practical knowledge and advice: offering emotional support; learning about self-help and making an active contribution.

Increased social competence and wellbeing was mentioned in almost every interview—this theme was derived from and included a number of sub themes which were labelled: the importance of social connections, developing and practicing skills, being competent helping others, gaining social approval, being valued and matching with others.

Meaning making and psychological wellbeing was mentioned in almost every interview—this theme was derived from and included a number of sub themes which were labelled: being productive and getting feedback, the importance of giving back, understanding own needs, sponsorship bridge to normal living, balancing commitments, identity changes.

### 2.1. Practical Knowledge and Self-Awareness

Information, knowledge, advice, and emotional support was typically provided from the sponsor to their sponsee. However, in addition to this expected directional flow, those providing sponsorship often reflected upon times when they had learnt valuable lessons from the person they were sponsoring. Reciprocal learning opportunities and discussions about benefits in the form of practical knowledge were common. However, these exchanges also included the sponsor learning about the strategies newer and less experienced members used to introduce structure into their life [14], in order to manage triggers and avoid relapse. Sponsors reported that this enabled them to maintain their abstinence as they often implemented the strategies that they learnt from the person they sponsored as they continued to practice their own programme’s steps. Sponsors also learned about what not to do in relation to their own recovery by observing and reflecting on the failures of less experienced members.

“*It’s about other people’s tricks that have worked for them-it’s on the same basis A will say something and I’ll say that’s never going to work for me and B will say something and I might think some of that might work for me, the whole thing might not work for me but I can pick a little bit of that out*” (Billy, 31 years old, 10 years in NA).

“*you’re also learning sometimes what you should not do, cause other people have done it and it will lead to this…do you know what I mean*” (Scott, 28 years old, 7 years in NA)

The themes relating to the importance of increased self-awareness and self-understanding in promoting the sponsor’s own abstinence were discussed by almost every participant, and in particular by participants who were newer to the process of providing sponsorship. However, during interviews, the majority of longer-term members also identified that the potential for developing practical knowledge whilst supporting others was increased for those who sponsored more than one other person. This was due to the multitude of other varied recovery experiences that they could be part of. Alongside the practical knowledge that was accrued whilst providing sponsorship, participants also discussed how sharing their personal stories with the person they are sponsoring provided them with the opportunity to engage in a deeper reflective process about their own experiences. Each time participants shared an experience, they were reminded of their addiction ‘as it was’, but they also were able to perceive it again with a newly developed appreciation, brought about by the ongoing process of sponsorship. Within this, participants described achieving a greater level of self-understanding and self-awareness by engaging with and supporting others. The participants below are discussing how they utilised the sponsorship role in relation to increasing their own self-understanding, the therapeutic value of the process and then how they benefitted from the process.

“*I took advantage, I used both to me advantage, supporting you know the people and using my past experiences and opening up with my past experiences to support them as well do you get what I mean. I was supporting them (…) I was co-counselling them but I was doing it myself (…) cause I was opening up and talking about my past experiences and explaining my likeness and how I can see where they are coming from my own experiences. So I was doing both, one them and myself by opening up*” (Stephen, 36 years old, 2 years in NA).

“*it’s difficult to describe because it is such a valuable tool of recovery it really is” (…) It helped me to see myself in a different way, so now whatever happens outside I kind of look to myself for the answers (…) you can change yourself and it has taught me to look to myself more.*” (Kelly, 26 years old, 1 year in AA).

During interviews, the majority of sponsors asserted the idea that the interpersonal benefit they drew from providing sponsorship to others had to be balanced against their own recovery needs. Those participants who were sponsoring more than one person as well as many of the women who reported additional child caring responsibilities tended to highlight the practicalities of time (commitments to work, family and leisure time) and the emotional energy it took to engage with and support newer members. There was a sense that whilst providing sponsorship was ‘good’ for the sponsor, they also recognised-they could over burden themselves supporting others, resulting in ‘burnout’. However, rather than highlight the potential for emotional strain, sponsors typically spoke about their ability to balance their own needs alongside sponsorship as evidence of their increased self-awareness.

### 2.2. Increased Social Competence and Status

Participants reported that they had developed social competences over time by sponsoring and helping others and that they benefitted from feelings of increased confidence within social and interpersonal relationships from doing so. In self-help groups, sponsors need a level of initial social competence and status to firstly attract others towards them but then also to maintain relationships with those they are helping over time. Initial social competence and in-group status is built up incrementally by the sponsor as a result of positively engaging and interacting with others in the group setting and, later, in one-to-one interactions in other settings. Sponsors often spoke during interviews about developing social competence in relation to their new abilities to engage with others in positive relationships, get on well with others, to maintain close relationships and to handle social interactions effectively [30]. Those who were able to engage in sponsorship with others and practice these skills over longer periods of time reported that they were also able to deepen these social competences so that it had potential benefits in other social contexts. Participants who reported high levels of social exclusion whilst using (such as being incarcerated or street homeless) such as Paddy, an active sponsor who spent a significant amount of time in prison and as a homeless drug user recalled

“*Enabling someone to warm to you (…). Learn to be open (…) not controlling or dictating to people, that you must do this do this, you must do that (…) for real…….it’s something that develops that is about openness again and being self-aware, but also aware of your own control issues*” (Paddy, 38 years old NA)

In terms of social processes, it is important to recognise that the offer of sponsorship cannot be made, sponsorship must be requested by the person who is seeking a sponsor and the request to be another’s sponsor was often experienced as a form of positive social approval [14]. Being asked to sponsor was also seen as a social and psychologically significant event in its own right. Whilst only a minority of participants discussed this specific theme of during interview, those who did explored it at length and placed great emphasis upon the experience. Below, Paddy and Anne discuss the process of being asked to sponsor for the first time and the sense of social validation and psychological wellbeing that they derived from the process:

“*It was like a badge of honour being asked, it was about the recognition from others, like an endorsement if you like, that I was doing something right*” (Anne, 48 years old, 2 years in AA).

“*I felt good, I felt honoured, I felt nervous (reflecting) I suppose it’s that responsibility which I’ve avoided all my life*” (Paddy, 38 years old, 10 years in NA).

The overwhelming majority of participants discussed the themes of sponsor competence, skills and abilities during interviews. Sponsors need to have specific skills in relation to understanding how self-help groups function [3]. Facilitating self-help processes such as step work requires sponsors to also be skilful at supporting others in the resolution of the others substance-related concerns [21]. These specialist skills are partly indigenous and accrued experientially from participating in groups; passed from peer to peer. Additionally, these skills are accrued from the ongoing extra-group process of being sponsored and providing sponsorship. In self-help it is usual for one individual to be sponsoring another and passing on specialist skills and assistance, whilst also being sponsored themselves. This arrangement, unlike most other 121 helping relationships, sees that the helper is also the helped. As such, the status achieved by providing sponsorship is always tempered by the vulnerability of receiving sponsorship. As Craig, 2 years abstinent, reports:

“*I have a sponsor, he has a sponsor, his sponsor is a sponsor and it’s a world-wide thing (…) there is a lineage of recovery. My sponsor is seven years, his sponsor is thirteen*” (Craig, 37 years old, 2 years in NA)

Those who provide sponsorship to others for an extended period of time, or within multiple sponsoring relationships, may go on to become known as “effective helpers”. This brings additional competence and the ongoing status that comes with it, providing an ongoing source of social approval for the sponsor. Only a small minority of sponsors outwardly discussed how they were valued by the person that they sponsored. Sponsors are expected to exercise humility in everything they do and in 12 step groups, having an elevated “ego” or elevated sense of self-importance is seen as dangerous to the individual’s recovery. As such, the majority of participants tended to remove themselves from the discussion and instead focus upon the respect they had for the skilfulness of their own sponsor. Indeed almost every participant discussed the theme of valuing their own sponsor. Jamie, a sponsor himself, reflects upon his appreciation of his own sponsor:

“*I value and believe what he says to me because he has more experience than what I have got. He has been in recovery a lot longer than me. They have gone through the process of the 12 steps and is applying them to his life, he’s still doing it today so when he says something to me I take it seriously*” (Jamie, 32, 4 years in NA).

The respect that Jamie and other sponsors held for their own sponsors was evident throughout the interviews. The participants looked up to their sponsor as a role model in how they have maintained their abstinence but also, importantly, in how they have provided effective sponsorship. This was a key theme and almost every participant discussed how they aspired to be as effective in their sponsorship of others; to constantly grow socially and gain recognition for their competence. At its essence, sponsorship provided an opportunity for the sponsor to be good at something ‘good’. More than simply refraining from using substances or engaging in anti-social behaviours, the sponsors were actively and purposefully supporting the recovery of others and this generated a great sense of personal achievement over time.

“*Aye recognition, recognition and at the same time having respect from peers, it is something that people never had before (…) you’ve got structure and a role to do…..you feel important*” (Brian, 42 years old, 10 years in NA).

“*… the feedback and the response* [from the person receiving the sponsorship] *is phenomenal*” (Helen, 42 years old, 3 years in NA).

Being in possession of specialist skills and being socially competent at engaging with others did not always equate to the sponsor being successful or competent at helping others maintain their own recovery. Sponsors did not always feel able to help others to resolve their substance-related concerns, they often made reference to their own experiences of being sponsored and reflected upon the common difficulty everyone had in identifying the ‘right’ sponsor. Of interest here is that whilst being the ‘wrong sponsor’ could be seen as a criticism wherein the sponsor lacked the ‘necessary skills’, the participants typically perceived this as a compatibility issue. Self-help members of 12 step groups are openly discouraged from being critical of their groups, group process and other ex or current members. This was discussed by a small number of sponsors as “taking another’s inventory”. However, the process of matching the right sponsor with the right sponsee was discussed by almost every participant as important but not always easy. Sponsors often discussed the social competence and skills required by them to enquire about the motivations, needs and expectations of the person asking for their sponsorship, and to say ‘no’ should this not match with their own.

### 2.3. Meaning Making and Psychological Wellbeing

Being a sponsor required participants to behave in ways which are congruent with the inherent values of recovery. Participants strive to achieve this congruence whereby they lived by key principles of humility, ‘giving back’ and being a productive member of society. Approximately half of those interviewed mentioned the importance of this theme during interviews and the ways in which sponsorship provided a mechanism through which to express these principles, providing a positive influence on the lives of others. Participants discussed how these behaviours and characteristics were now part of who they had become as individuals and how they were very different in relation to the characteristics which they had as “addicts” or when in “active addiction” [31,32]. Participants also reported that they gained an increased sense of meaning and purpose in life from providing sponsorship. Meaning making in self-help is derived from the personalised understanding individuals develop about their own recovery needs [33] and recognition of how important particular processes such as engaging in helping others are in facilitating the process [34]. In a very practical way, sponsors were able to plan for their future and longer-term recovery because they knew they had access to non-using relationships, in which they could practice their groups principles. Over time, these principles and practices associated with sponsorship become more significant and more important to the individual’s own journey.

“*It’s like where you have come from, running about the streets scoring and things, doing this and doing that, like I said to this day, it [sponsorship] gives people a purpose and a responsibility that they have never had before*” (Harry, 37 years old, 3 years in AA).

“*There is a set of principles what you implement in your life which enables you to have a productive life and be a productive member of society*” (Craig, 27 years old, 2 years in NA).

Moreover, incongruence with these principles and practices presented a direct threat to the recovery of some individuals.

“*If I don’t live by these principles I’ll end up going back and will use drugs you know*” (Craig, 27 years old, 4 years in NA).

Over half of those who were interviewed recognised that sponsoring relationships provided an important relational space within which to practice new identities and productive lives in the wider self-help community. Sponsorship also aided the extension of meaning making into wider aspects of their lives. The relationships sponsors engage in are based on shared experiences, the shared goal of recovery and mutual recognition and acquaintance. However, over the longer term, it is not unusual for sponsors to develop a friendship group with those who they were sponsored by or who they provided sponsoring to. This particular theme was discussed by the small number of longer-term sponsors who were also committed to the continuation of their groups and involvement in self-help.

“*You accumulate life as you go on in recovery, life in recovery is a bridge to normal living and I have accumulated other networks. I’ve also got someone (…) who is my sponsor and he has been for many years. I have opened up to him, I’ve told him some stuff, I have now a really good relationship with him*” (Paddy 38 years old, 10 years in NA)

As sponsors moved forward to achieve what they described as a more productive life outside of the self-help community, the social and psychological significance of sponsoring lessened. Approximately one-quarter of participants discussed this theme and, for these individuals, meaning in life is mostly generated from activities outside of the group and sponsorship is performed out of a sense of duty and obligation. Those who move into more conventional social networks do continue to recognise the importance of sponsorship within the self-help community, but they no longer feel the expressed need to sponsor to generate meaning. As they moved away from the self-help groups and into more conventional types of social relationships and networks, the meaning that was generated from sponsoring was often replaced by positive social interactions with peers from outside the self-help community, or from activities such as employment, family or leisure. More often than not, their involvement in self-help and sponsoring is continued, driven by the belief that they will always be in a form of recovery and that they should maintain contact and be active ‘just in case’.

“*It’s a bit of a balancing act (…) I just go with the flow sort of thing. I’ll go through the stage where I go down to one sponsee and then someone else will ask me which I totally understand. But I try not to really think about it, because as soon as I think about doing this stuff for the rest of my life and sponsoring people, that’s when I think I don’t want to do it at all*” (Damien, 31 years old, 7 years in NA)

## 3. Discussion

This study has shown that people who provide sponsorship to others in recovery derive a significant range of psychological and social benefits. These benefits are wide ranging and are key to the ongoing recovery of the sponsors themselves. They include increased self-awareness, increased psychological wellbeing, social competence, positive social approval and status, meaning making and practical access to non-using social relationships and contexts. Sponsorship is a complex multidimensional and socially mediated process, which runs as an adjunct to and compliments general 12 step self-help participation and processes. Like others in similar fields (mental health), we have identified that the attribute of increased self-awareness and self-understanding derived from sponsoring others is key in attaining and sustaining abstinence and recovery over the short and longer terms [15,16,17,34]. However, our study has also found that the social competence that individuals accrue and develop by engaging with others in sponsorship and their ability to find purpose and meaning in their sponsorship activities were equally important parts of the recovery process. The process of providing sponsorship improves the quality of social relationships between all self-help users by providing individuals with a sense of how to behave in groups and different forms of relationships [35]. The sponsor is also provided with a social context for re-socialisation, wherein they have the opportunity to connect with traditional values and beliefs about helping others and being seen to be a more productive member of their group and society. As a social process, the practice and development of key social skills in sponsorship are particularly beneficial to the recovery of those who were withdrawn, disenfranchised or distanced from conventional societal beliefs and values prior to accessing self-help [22,36]. The social approval that is derived from being recognised and asked to sponsor is also more beneficial for those who had experienced significant forms of substance use, social isolation and/or felt like they have been a burden to others [37] prior to their involvement in self-help. Over time, the sense of psychological wellbeing that is derived from successfully helping others in self-help is increased, away from and in addition to that experienced in the individual’s main group. This is largely because of the increased significance of the sponsoring relationship and the intimacy and commitment to action which underpinned this type relationship. In this context, helping others, being productive and taking responsibility are meaningful practices but they are also key signifiers [32] in the self-categorisation process, as the self-help user transition into a new “non-addict” identity [12,13]. Over the longer term, sponsorship enables sponsors to practice a form of conventional living whilst still being part of a self-help group and maybe beneficial for those with limited access to conventional social worlds, groups and relationships. Sponsorship provides the sponsor with social opportunities to enhance and maintain the new and more salient “in recovery” identity they are developing from their involvement [12,13].

This research provides a relational perspective upon the recovery process and the ways in which social and psychological benefits are derived for the individuals who provide sponsorship. The individual enters self-help and develops competence in engaging with others, self-help and self-help processes. They then use their elevated social status as a sponsor to attract others towards them in relationships on mutual recognition and acquaintance [21]. Over time, those sponsoring build and use their social competence to help others and in turn gain a form of in-group positive social approval and psychological wellbeing. Our findings give support to the work of others which have found that sponsors have elevated in-group “senior status” [3] and are seen as having “expertise” [31] in self-help processes, helping others and the resolution of their own substance-related concerns. However, rather than simply endorsing these types of ideas, our research has also explored the ways in which those who provide sponsorship, utilise their social position, status and the competences that they develop to work towards resolving their own recovery related concerns.

In terms of limitations, this is a small-scale qualitative interview based-study, with a limited sample in relation to diversity and settings. This type of research does have some limitations associated with validity, applicability and generalisability to other settings and contexts [38]. It is also limited in its focus because all of those sampled and interviewed had generally benefitted over the short and longer terms from their involvement in self-help groups and in providing sponsorship to another member/members. In this context, it would have been useful to interview those who either refused to sponsor others or those who disengage with self-help groups because of sponsorship. This type of focus would have been made difficult, but not impossible, by the fact that individuals in groups such as NA and AA are openly and actively discouraged from being negative or critical of their group and or other members. In taking up a focus on sponsorship processes alone, this study does not consider the statistical significance of each of the benefits in relation to the individual users, rather it focusses on identifying the interactions or cumulative affects relating to the benefits of supporting others in the wider group context. Despite these limitations, this study, like others in different fields, has gone some way towards giving a voice to the “wider lived experiences” of users as they engage in self-help groups and self-help processes with each other [24,25,26].

Our findings have implications for public health and health and social care professionals working with individuals who use substances. Whilst supporting an individual to attend a self-help group might promote abstinence, encouraging a recovering user to sponsor another person’s recovery may further benefit their own social and psychological wellbeing. Users of self-help are more likely to benefit over the short and longer terms by engaging in processes such as helping others and by providing sponsorship. Users of self-help themselves also need to recognise the implication that recovery is an incremental process. Recovery requires individual action on their own behalf, self-application and self-determination but also virtual and actual benefits that can be accrued if they take an interest and make a commitment to getting involved in helping with the recovery of others. In terms of future research, those who work with individuals in recovery and self-help groups need to consider the implication of this research and a number of theoretical and practical concerns. The first relates to those who study in the field of recovery and the need to move away from the narrow definition of abstinence that is often used to define recovery and the more practical assumption that individuals are helped towards recovery by entering a group, engaging with the group’s programme of change and applying group principles in their lives. What is needed here is a recognition that self-help groups are essentially social microcosms and that more dynamic or integrated ways are needed to explore and explain individual functioning and social processes over time as individuals come together with others to resolve their substance-related concerns. Future researchers also need to recognise that opportunities still exist to explore the ways in which specific components or aspects of self-help groups are experienced and how they relate to/impact on the recovery process [39].

## 4. Conclusions

The recovery process involves achieving sustained abstinence from alcohol and drugs as well as the individual developing the personal, social, and psychological skills which support a transition into a non-addict identity. The provision of sponsorship to others plays an important role in supporting the functioning of self-help groups and the recovery of the individual providing it. This research has illustrated that recovery is a dynamic process and complex interplay between the development, refinement and acknowledgement of knowledge and skill and the meaning-making potential of providing sponsorship to support the sponsor’s social and psychological recovery. Sponsorship is a key transitional process that facilitates the sponsor’s transition into non-using social groups in a way that maintains the sponsor’s sense of self as a congruent and accepted social group member.

## Data Availability

Due to the nature of this research, participants of this study did not agree for their data to be shared publicly, so supporting data is not available.

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
