# Peer review of "An Exploration of the Psycho-Social Benefits of Providing Sponsorship and Supporting Others in Traditional 12 Step, Self-Help Groups"

_ijerph, 2021, doi:10.3390/ijerph18052208_

Round 1

Reviewer 1 Report

This manuscript reports on a unique qualitative investigation of the social and psychological benefits of being a sponsor in a 12-step substance use recovery program. The paper presents a rich dataset of interviews from 36 former alcohol and heroin-crack users who have recovered with the help of AA or NA and become sponsors for those starting their recovery. The paper describes the ways in which being a sponsor has benefited these participants socially and psychologically but does not meaningfully report on challenges that sponsoring caused sponsors, or comment on how sponsoring directly impacted alcohol/drug abstinence. The paper provides much needed insight into the benefits of sponsoring and the social functioning of AA/NA groups. One of my primary concerns was that it appeared the authors did not extract themes from their qualitative dataset in an exploratory fashion, but instead found three themes that fit with Bourdieu’s typology of capitals, limiting the information we can derive from this dataset to only information that conforms to this theory. I presume this is the reason that other themes (e.g., challenges to one’s own abstinence) that participants experienced as a result of sponsoring are not included in the results. It is unclear why the authors would choose to take this approach rather than exploring all present themes, given the lack of previous research on the impact of sponsoring to the sponsor and the high value that such an analysis would have for the field.

Abstract:

The authors conclude “Sponsorship is a process which is most beneficial for those who have experienced high levels of substance related harm prior to accessing self-help and/or those who have little access to wider social networks”. I hesitate to agree whether this statement is supported by the research reported in this article, because it is unknown whether the people in the present study experienced higher substance-related harm/less access to social networks than the average person in recovery (i.e., the sample here may not be representative of all people in recovery).

Introduction:

Bourdieu’s typology of capitals is introduced at the end of the introduction as a theory that guided the study’s aims, but there is no review of the research linking this theory to recovery or addictive processes. I am also a bit unclear on how these typologies are linked to the study’s design and data analysis: were themes extracted from the interviews with these typologies as a guide?

There is an attempt to also address the conflicting definitions of ‘recovery’ in the introduction. As written, this feels tangential to the aims of the paper, which aims to describe the social and psychological benefits of being a sponsor. Perhaps the authors include this in order to deflect focus from abstinence/use outcomes (as I noted are missing) and focus on social/psychological recovery instead. Perhaps this would come across more clearly if it were more thoroughly integrated throughout the introduction and discussion, and if the authors introduced Bourdieu’s typologies earlier, focusing on how these components are an important part of the recovery process. In general, more integration is needed to tie together all of these concepts.

Method:

Can the authors briefly describe what a thematic semi-structured topic guide is?

Did the authors collect data on how long ago each participants’ sponsoring experience was? I am wondering, were some participants reflecting upon sponsorship without having engaged in sponsorship for several years?

The data analysis section states that the data were “coded”, but is very vague about what this means. Were the data analyzed in audiovisual format, or just audio? Were the data being analyzed for subject/content only? Were the subjects the same themes that were initially presented in the method, or were additional themes identified? What are some examples of these themes? Can you report the interrater reliability coefficient on double-coded data?

Did the authors extract themes and then map them onto Bourdieu’s typology of capitals, rather than deriving themes organically from the dataset? If so, this approach is more ‘confirmatory’ than ‘exploratory’, and this should be made clearer in the intro/aims, method, and results. My enthusiasm for the paper is also dampened if this was the case—an exploration of all themes has not been done before and I am very curious about what other themes emerged.

Results & Discussion:

After reading the results, I am unsure if the authors presented all the themes. Can the authors add a short descriptive section describing how many themes were derived from the interviews, what they were, and how common each theme was among the participants interviewed?

In the discussion, the authors infer that the benefits conferred by sponsorship (self-understanding and self-awareness) may help sponsors stay abstinent—did the sponsors themselves say anything about this? These results would be very interesting to include in the paper, noting that of course abstinence is only one of the many important components to recovery.

Minor comments:

  • Lines 104-110: Have these longitudinal studies attempted to explain the temporal effect at play? It could be that those who become sponsors have an improved chance to sustain abstinence, or that those who sustain abstinence are likely to eventually become sponsors for others.
  • At the end of the introduction (lines 134-136), the authors state a bit of what they found in their study, but I would usually only expect this information to be shared in the results and discussion.
  • Line 423, I can’t follow the sentence: “Rather than simply identifying that sponsors do have elevated status in groups and expertise in self-help, this research has explored the ways in which those who provide sponsorship, utilise their social position, status and the competences they develop to work towards resolving their own recovery related concerns.” There are a few rather long sentences that are difficult to follow in other areas of the manuscript as well.

Author Response

Reviewer A Comments

Response/Amendments

Abstract

The authors conclude “Sponsorship is a process which is most beneficial for those who have experienced high levels of substance related harm prior to accessing self-help and/or those who have little access to wider social networks”. I hesitate to agree whether this statement is supported by the research reported in this article, because it is unknown whether the people in the present study experienced higher substance-related harm/less access to social networks than the average person in recovery (i.e., the sample here may not be representative of all people in recovery)

We appreciate reviewer A’s hesitation and have changed the wording of this sentence in the abstract and main body of the manuscript (see track changes) from what is was previously to…..

Additionally, sponsorship is a process which is beneficial for those who have little access to wider social networks.

Introduction:

Bourdieu’s typology of capitals is introduced at the end of the introduction as a theory that guided the study’s aims, but there is no review of the research linking this theory to recovery or addictive processes. I am also a bit unclear on how these typologies are linked to the study’s design and data analysis: were themes extracted from the interviews with these typologies as a guide?

There is an attempt to also address the conflicting definitions of ‘recovery’ in the introduction. As written, this feels tangential to the aims of the paper, which aims to describe the social and psychological benefits of being a sponsor. Perhaps the authors include this in order to deflect focus from abstinence/use outcomes (as I noted are missing) and focus on social/psychological recovery instead. Perhaps this would come across more clearly if it were more thoroughly integrated throughout the introduction and discussion, and if the authors introduced Bourdieu’s typologies earlier, focusing on how these components are an important part of the recovery process. In general, more integration is needed to tie together all of these concepts.

Method:

Can the authors briefly describe what a thematic semi-structured topic guide is?

Did the authors collect data on how long ago each participants’ sponsoring experience was? I am wondering, were some participants reflecting upon sponsorship without having engaged in sponsorship for several years?

The data analysis section states that the data were “coded”, but is very vague about what this means. Were the data analyzed in audio visual format, or just audio? Were the data being analyzed for subject/content only? Were the subjects the same themes that were initially presented in the method, or were additional themes identified? What are some examples of these themes? Can you report the interrater reliability coefficient on double-coded data?

Did the authors extract themes and then map them onto Bourdieu’s typology of capitals, rather than deriving themes organically from the dataset? If so, this approach is more ‘confirmatory’ than ‘exploratory’, and this should be made clearer in the intro/aims, method, and results. My enthusiasm for the paper is also dampened if this was the case—an exploration of all themes has not been done before and I am very curious about what other themes emerged.

Response

We would like to thank reviewer A for their reflections and comments here. 

Bourdieu’s concepts of capital do not appear in the research design and were not used as a framework (also see next section) to analyse data.  Its inclusion in the introduction complicates the manuscript. Therefore, we have removed reference to Bourdieu’s concepts from the introduction section.  We have also rewritten the small part of the discussion section.  Here we have replaced reference to Bourdieu grand theory and spoke about our research and theoretical contribution in relation to empirical theorists like Smith (1997) and Yeung (2007).   

We have changed this to now read  topic guide and provided a description it now reads: A thematic topic guide which contained a set of standard questions and prompts to facilitate discussion was developed and informed the basis for the semi structured interviews.  The topic guide questions which related to the following areas: group history and involvement, sponsorship provision in group, motivations to sponsor, experiences of sponsorship, perceived benefits and actual impacts, level of planned involvement in sponsorship over time

The minimum amount of time anyone had been sponsoring was 3 months and the maximum was 10 years. The average amount of time spent sponsoring others was 3-4, this was gauged as an estimate during interview.  This information is detailed within the participants section lines 178-181

We thank the reviewers for the comments here and have rewritten this section for clarity. We have responded by expanding the discussion significantly on the theory, methods used and the description of the processes we undertook in data analysis.  We have also given examples of initial and then later codes/themes that emerged as the study and data analysis progressed.  We also report on the processes that informed the second coding of the data and the meetings which occurred in relation to ensure reliability.

Results & Discussion:

After reading the results, I am unsure if the authors presented all the themes. Can the authors add a short descriptive section describing how many themes were derived from the interviews, what they were, and how common each theme was among the participants interviewed?

In the discussion, the authors infer that the benefits conferred by sponsorship (self-understanding and self-awareness) may help sponsors stay abstinent—did the sponsors themselves say anything about this? These results would be very interesting to include in the paper, noting that of course abstinence is only one of the many important components to recovery.

All of the themes are reported on. We have clarified this in each of the results sections by adding more detail in relation how significant each of the themes was among the participant population.  

We have explicit reference within the results to the benefit of self-awareness and self. These are: 

Sponsors reported that this enabled them to maintain their abstinence as they often implemented the strategies that they learnt from the person they sponsored as they continued to practice their own programme’s steps.

The themes relating to the importance of increased self- awareness and self-understanding in promoting the sponsoring own abstinence were discussed by almost every participant, and in particular by participants who were newer to the process of providing sponsorship.

Minor comments:

Lines 104-110: Have these longitudinal studies attempted to explain the temporal effect at play? It could be that those who become sponsors have an improved chance to sustain abstinence, or that those who sustain abstinence are likely to eventually become sponsors for others.

At the end of the introduction (lines 134-136), the authors state a bit of what they found in their study, but I would usually only expect this information to be shared in the results and discussion.

Line 423, I can’t follow the sentence: “Rather than simply identifying that sponsors do have elevated status in groups and expertise in self-help, this research has explored the ways in which those who provide sponsorship, utilise their social position, status and the competences they develop to work towards resolving their own recovery related concerns.” There are a few rather long sentences that are difficult to follow in other areas of the manuscript as well.

This is an interesting point raised by the reviewer. The studies did not examine this unfortunately. We have added this limitation to the introduction section (line 110-112)

We were following guidelines provided by the journal in relation to this addition but are happy for it to be removed if the editor decides so. 

We have reworded this sentence:

However, rather than simply endorsing these types of ideas, our research has also explored the ways in which those who provide sponsorship, utilise their social position, status and the competences they develop to work towards resolving their own recovery related concerns.

 We have also restructured multiple overlong sentences throughout the document.

Author Response

Reviewer B comments

There are a couple of minor to moderate weaknesses that should be addressed before publication. One is a discussion of how themes were identified during the qualitative analysis process

We thank the reviewers for the comments here and have rewritten this section for clarity. We have responded by expanding the discussion significantly on the theory, methods used and the description of the processes we undertook in data analysis.  We have also given examples of initial and then later codes/themes that emerged as the study and data analysis progressed.  We also report on the processes that informed the second coding of the data and the meetings which occurred in relation to ensure reliability.

Another is a need for a discussion of Bourdieu’s work, especially in regard to the typology of capitals: symbolic, social and cultural. How did he use this typology and what research problem and population did he address? The authors allude to his research in different parts of the manuscript but should consider concentrating the discussion in the introduction, when Bourdieu is introduced to the reader for the first time.

Thank you for this.  In response to this and reviewer A’s comment that Bourdieu framework was not central to the research design we have removed reference and replaced with reference to a more relevant and empirical.

Round 2

Reviewer 1 Report

The authors made many revisions to their manuscript to address the concerns raised in the previous review. Many of these concerns were clearly addressed and have been resolved, particularly those surrounding the use of Bourdieu’s typology of capitals. However, some original concerns were not addressed.

The authors added a new conclusion statement to the introduction, but the statement makes broad conclusions about the recovery process, something that was not really examined in this study. I wonder if a more suitable conclusion would comment in a more focused way on the impact of being a sponsor on social and psychological well-being (and perhaps make an extension to how this impacts recovery).  

Were the authors unable to calculate a coefficient of interrater agreement, such as kappa, or even just a percentage? Without a number, the authors’ judgment that the rate of agreement was “high” is subjective and we cannot know if they agreed 60% of the time or 90% of the time.

When I asked for a descriptive section at the beginning of the results, I was looking for a descriptive statistics/results overview section, which gives a broad overview of your results section before diving into the first theme. Something like “We extracted 3 themes: A, B, and C. A was the primary theme in 50% of interviews, B was the primary theme in 30% of interviews, etc.” Also valuable: “A was discussed in 95% of interviews, B was discussed in 80% of interviews, and C was discussed in 50%”.  Just a broad overview so the reader can understand what the data looked like as whole.

It feels unsatisfying to read that there were only three themes and these three themes were present in most interviews. How is it possible that these sponsors had such homogeneous experiences? Perhaps the themes are too broad. Indeed, within the last ‘theme’, the authors identify several sub-themes that are endorsed at different rates… Can the authors make titles for ‘sub-themes’ and add them to the descriptive statistics/results overview section?

Author Response

We would like to extend our thanks to reviewer A and reviewer B for their time in committing to this paper and reviewer B for their swift response in assessing and reviewing. 

Reviewer A

Comments/Responses

The authors added a new conclusion statement to the introduction, but the statement makes broad conclusions about the recovery process, something that was not really examined in this study. I wonder if a more suitable conclusion would comment in a more focused way on the impact of being a sponsor on social and psychological well-being (and perhaps make an extension to how this impacts recovery).  

We have now changed this to:

Sponsorship draw a range of benefits from supporting others, this includes, increased self-awareness and self-understanding, increased social competence, positive social approval and psychological wellbeing. Over the longer term sponsorship provides opportunities for those providing it to be productive, make meaning and maintain a non-addicted identity.

Were the authors unable to calculate a coefficient of interrater agreement, such as kappa, or even just a percentage? Without a number, the authors’ judgment that the rate of agreement was “high” is subjective and we cannot know if they agreed 60% of the time or 90% of the time.

When I asked for a descriptive section at the beginning of the results, I was looking for a descriptive statistics/results overview section, which gives a broad overview of your results section before diving into the first theme. Something like “We extracted 3 themes: A, B, and C. A was the primary theme in 50% of interviews, B was the primary theme in 30% of interviews, etc.” Also valuable: “A was discussed in 95% of interviews, B was discussed in 80% of interviews, and C was discussed in 50%”.  Just a broad overview so the reader can understand what the data looked like as whole

Because of the subjectivity of the process it would not have been appropriate to calculate a co-efficiency of the interrater agreement.  However. We have provided a more in-depth explanation of the data analysis process in the analysis section:  

Those members of the research team facilitating the secondary coding were given an audio recording of interviews, a word version of each of the transcript and blank excel sheet which contained all of the existing codes.  No new codes were added or deemed necessary at the conclusion of the secondary coding phase. In addition to this and after in-depth discussion there was universal agreement among all three researchers in relation to the primary themes and sub codes. 

In response to providing a broad overview of the data and its significance we have now added a descriptive overview of the data into the results section. In this additional section we identify the main themes and we label the sub themes as requested.  As this is a qualitative study with aims to explore the experiences of self-help users.  We did not map statistical data or a percentage overview relating to each of the individual sub themes in the form requested by reviewer A. However, in response to the reviewers comments we have provided detail and clarity on the main and sub themes.  We have recognised in the results sections that each of the main themes were discussed almost universally by respondents.  This recognition is in addition to the previous changes we made to the results section, in which reflected on how significantly some of the sub themes were discussed by and in relation to sub groups of respondents in the interviews.     

It feels unsatisfying to read that there were only three themes and these three themes were present in most interviews. How is it possible that these sponsors had such homogeneous experiences? Perhaps the themes are too broad. Indeed, within the last ‘theme’, the authors identify several sub-themes that are endorsed at different rates… Can the authors make titles for ‘sub-themes’ and add them to the descriptive statistics/results overview section?

See above but also: we hope, given the minor amendment recommendation of the last review that the changes above will suffice but we have also considered the point made by reviewer A and reworded part of the limitations section to recognise the lack of statistical content analysis in this study.

In taking up a focus on sponsorship processes alone, this study does not consider the statistical significance of each of the benefits in relation to the individual users, rather it focusses on identifying the interactions or cumulative affects relating to the benefits of supporting others in the wider group context.